# The Modulating Effect of Ethanol on the Morphology of a Zr-Based Metal–Organic Framework at Room Temperature in a Cosolvent System

**Yunzhuo Li [1], Zirong Tang [2] and Chen Chen [2,*]**

[1]  Wuhan Britain-China School, Wuhan 430074, China; liyunzhuotr@gmail.com
[2]  School of Mechanical Science and Engineering, Huazhong University of Science and Technology, Wuhan 430074, China; zirong@hust.edu.cn
*   Correspondence: chenchen_@hust.edu.cn

**Abstract:** We report that ethanol, used together with water, plays a crucial role in tuning the structures of a zirconium-based metal–organic framework and the 12-connected MOF-801, as well as the possible mechanisms of this modulating effect. By employing a cosolvent system of ethanol and water at just under room temperature without the presence of a monotopic carboxylic acid as the modulator, MOF-801 in various morphologies of different sizes could be synthesized. A linear correlation between the ethanol/water ratio and the crystal sizes is also demonstrated. The growth mechanism is mainly explained by ethanol's bonding with the metal ion clusters and the Marangoni flow effect. Ethanol competes with the linker molecules in coordinating with the Zr metal clusters, a role similar to that of the modulators. The Marangoni flow effect, which dominates at a certain solvent ratio, further promotes the 1D alignment of the MOF-801 crystals.

**Keywords:** metal–organic framework; MOF-801; modulators; ethanol; Marangoni flow effect





## 1. Introduction

Metal–organic frameworks (MOFs), a group of porous crystalline materials, have raised a few eyebrows due to their high surface area [1]. The permanent porosity of the MOFs allows gas molecules to move in and out without barriers, making them a promising material for gas absorption and storage [2,3]. Among all the MOFs that have been synthesized, Zr-based ones have a proper place due to their large family ranging from 6-connected to 12-connected. MOF-801 is a Zr-based MOF that stands out prominently for its considerable water absorption capability at atmospheric pressure and room temperature [4]. It encompasses 12-connected $Zr_6O_4(OH)_4$ secondary building units (SBUs) and linear ditopic fumaric acid linkers. The low reversibility of Zr–O bonds makes the synthesis of high crystallinity challenging [1]. Zr-based MOFs are commonly synthesized by reacting an appropriate metal source with organic linker molecules dissolved in an organic amide solvent with a certain modulator (typically monocarboxylic acids such as formic, acetic, or benzoic acid) at temperatures ranging from 60 to 140 °C for several hours or several days [1,5]. The MOFs synthesized without the presence of a modulator are often amorphous or of poor crystallinity [6]. The modulator molecules bond with the SBUs formed in situ at certain positions, blocking the extension of the structure in these specific directions, allowing polyhedron MOF crystals with higher crystallinity to form [7]. Schaate et al. reported that the Zr-based metal–organic frameworks obtained by exclusively mixing the water solution of the metal source and fumaric acid without modulators, however, is amorphous with disappointingly low crystallinity, as the water increases the reticulation rate [8]. By using two solvents or a cosolvent system, researchers may tune the structures of the MOFs. By changing the ratio between ethanol and water in the synthesis, Jang et al. were able to synthesize ZIF-67 of various shapes and sizes [9]. However, the growth of

12-connected MOF-801 is a different story from that of the single-atom metal nodes. The formation of highly crystalline Zr-based MOFs is rather challenging, limiting the scale of their mass production. Dai et al. came up with a method to synthesize Zr-based MOFs, but it still requires the addition of monotopic modulators [10]. The Marangoni flow effect is the convection between liquids up the surface tension gradient [11]. When the solvent of low surface tension evaporates, that of higher surface tension simultaneously gathers and condenses near the surface, forming a drag force which promotes one-directional flow from the low-surface-tension solvent to the high-surface-tension solvent. The Marangoni flow effect is known to be able to direct the one-directional movement of nanoparticles, regulating the patterns of these particles [11,12]. We herein report the role of ethanol in the morphology of MOF-801, whose structure can be tuned at room temperature by using a water–ethanol cosolvent system, in addition to its mechanism during the reticulation. Ethanol, just like fumaric acid, may bond with the metal clusters, serving a purpose similar to that of the modulators. The Marangoni flow effect also has a significant impact at specific solvent ratios. However, outside of this ratio, the nanocrystals are too small and far apart or they interconnect.

## 2. Materials and Methods

Zirconium sulfate ($Zr(SO_4)_2$, AR, 100 g) and fumaric acid($C_4H_4O_4$, AR, 250g) were purchased from Shanghai Zhanyun Co. Ltd. (Shanghai, China). Ethanol ($C_2H_5OH$, AR, 500 mL) was purchased from XILONG SCIENTIFIC Co. Ltd. (Shantou, China). All materials were used without further purification.

First, 1.78 g of zirconium sulfate was dissolved in 20 mL of water and 0.59 g of fumaric acid was dissolved in 20 mL of ethanol [13,14]. These two solutions were gently mixed using a dropping pipette. The mixture was maintained still at room temperature for 30 min to allow nucleation and then stirred for 240 min (4 h). The white powder of MOF-801 was obtained by centrifugation (9000 rpm) for 10 min. The obtained sample was washed thrice with a blend of 20 mL of water and 20 mL of ethanol and twice with pure ethanol. Then, it was dried at 60 °C for 3 h. This sample was labeled 'A'.

Then, ceteris paribus experiments were carried out. The masses of zirconium sulfate and fumaric acid were fixed, while the ratio between ethanol and water was changed to 2, 3, 4, 0.5, 0.33 and 0.25, labeled as samples B, C, D, E, F and G, respectively. We first fixed the volume of water at 20 mL and changed the volume of the ethanol added to 40, 60 and 80 mL, as shown in Table 1. Then, we fixed the volume of ethanol and increased the volume of water added to 40, 60 and 80 mL. At last, the above procedures were duplicated on each occasion to obtain MOF-801 crystals.

**Table 1.** Conditions used to obtain the MOF-801 samples.

| Sample | A | B | C | D | E | F | G |
|---|---|---|---|---|---|---|---|
| Ethanol: Water | 20 mL:20 mL | 40 mL:20 mL | 60 mL:20 mL | 80 mL:20 mL | 20 mL:40 mL | 20 mL:60 mL | 20 mL:80 mL |

All the samples prepared were then sent for characterization. The SEM tests were carried out on a JEOL JCM-6000 scanning electron microscope. Powder X-Ray diffraction (PXRD) was performed on a Bruker AXS D8 (Bruker, Karlsruher, Germany) with copper as the radiation source. The compositions of the materials were determined by X-ray photon spectroscopy on an Omicron Dar400 (OMICRON, Taunusstein, Germany) with an achromatic aluminum X-ray source. The specific functional groups present in the material were investigated by FT-IR on a PerkinElmer Frontier (PerkinElmer, MA, USA).

## 3. Results and Discussion

The various morphologies of the MOF-801 samples, as well as their PXRD patterns, are shown in Figure 1. The morphology of MOF-801 crystals formed in 20 mL of water and 20 mL of ethanol resembled polyhedrons sized approximately 100 nm.

When the ratio between ethanol and water was changed to 2, the solution became turbid more slowly, indicating a lower reticulation rate. The samples obtained had a rod morphology. When the ratios of water to ethanol increased to 20 mL:60 mL and 20 mL:80 mL, relatively large polyhedral granules sized about 300 and 500 nm were obtained. The XRD analysis shown in Figure 1 shows the phase-pure synthesis of the MOF-801 crystals [4,14].

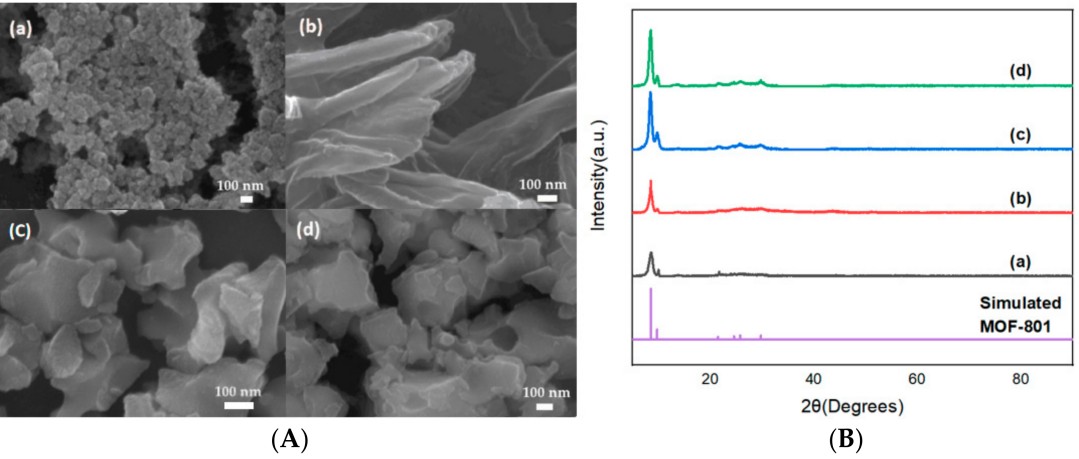

**Figure 1.** SEM (**A**) and XRD (**B**) diagrams of the samples prepared in (**a**) 20 mL of water, 20 mL of ethanol, with polyhedrons of approximately 100 nm; (**b**) 20 mL of water, 40 mL of ethanol, with rods; (**c**) 20 mL of water, 60 mL of ethanol, with polyhedrons of approximately 300 nm; (**d**) 20 mL of water, 80 mL of ethanol, with large polyhedrons of approximately 500 nm.

Subsequently, the ratio of ethanol to water was decreased below 1. When the proportion of water in the reaction increased, the white precipitate formed much faster, indicating a higher rate of nucleation. When the volume of water was 40 mL, polyhedron granules sized approximately 40 nm were obtained. These granules began to show partial interconnection, with a small minority of the crystals accreting. Then, when the volume of water was increased to 60 and 80 mL, small nanosized polyhedrons of lengths less than 10 nm that were intergrown and could not be separated by ultrasound were observed, as shown in Figure 2. It should be noted in the XRD diagrams in both Figures 1 and 2 that, as the ratio between ethanol and water decreased, the peaks became lower and wider, indicating smaller crystal sizes of the MOF-801 crystals. This is coherent with our observations from the SEM that decreasing the ratio led to a fall in crystal size. Furthermore, we used the Debye–Scherrer equation to calculate the sizes of all these crystallites [15]. When the ratios of ethanol to water were 0.25, 0.33, 0.5, 1, 3 and 4, the corresponding crystal sizes were 9, 13, 39, 121, 294 and 536 nm, respectively, derived from the Debye–Scherrer equation. This supports our observation from the SEM.

The exception of the shape of the MOF-801 should be noted when the solvent ratio was 2. In order to understand the process of the growth of this kind of rod-like MOF crystal, we performed an ex situ analysis during the growth of the MOF-801 nanorods [9]. As shown in Figure 3, we were able to determine the phases that the MOF-801 crystals which underwent the reticulation process. We obtained samples under different reaction times, i.e., 15 s, 30 min and 3 h. When we analyzed the MOF-801 samples formed in 20 mL of water and 40 mL of ethanol in 15 s, we observed a one-directional alignment of thread-like crystals that were interconnected. Although the crystals were not separated from each other, the preferred growth direction of these crystals was identical. When the reticulation time was increased to 30 min, the one-directional alignment of the crystal growth could still be observed. The crystals grew toward the identical direction, but each single crystallite began to separate from each other. The interconnected structures began to collapse and single crystals started

to form. When the reaction proceeded for 4 h, the rod-shaped MOF-801 single crystals become independent from each other.

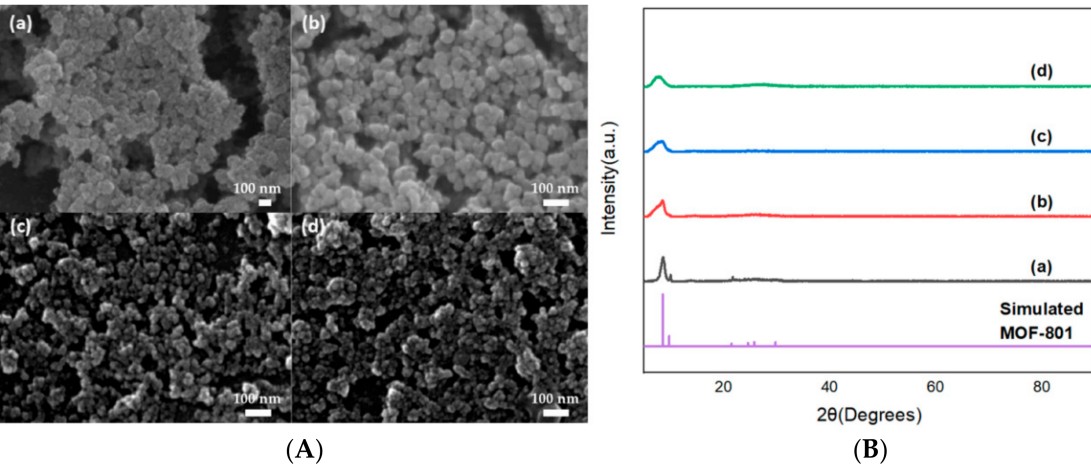

(A)  (B)

**Figure 2.** SEM (**A**) and XRD (**B**) diagrams of the samples prepared in (**a**) 20 mL of water, 20 mL of ethanol, with polyhedrons of approximately 100 nm; (**b**) 40 mL of water, 20 mL of ethanol, with granules of approximately 40 nm; (**c**) 60 mL of water, 20 mL of ethanol, with intergrown nanosized crystallites; (**d**) 80 mL of water, 20 mL of ethanol, with intergrown nanosized crystallites.

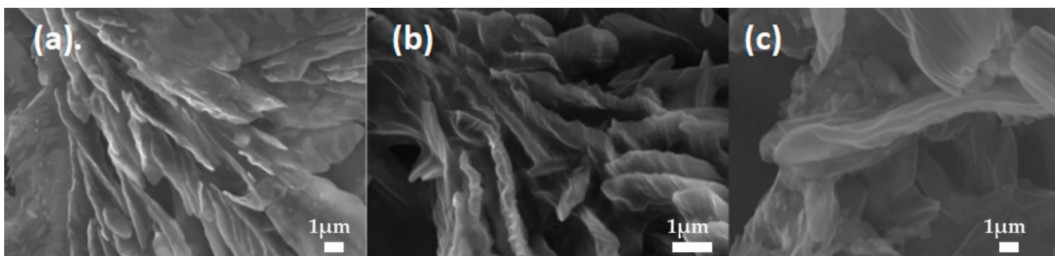

**Figure 3.** The ex situ analysis of the MOF-801 samples formed in 20 mL of water and 40 mL of ethanol after reticulation for (**a**) 15 s, (**b**) 30 min and (**c**) 3 h.

With the aim of determining the relationship between the volume ratio of ethanol to water and the crystal sizes, we used Excel to plot a best-fit line by linear fitting. The linear fitting graph is shown in Figure 4. The trend of the crystal size changing with the ratio seemed to be linear.

We then calculated the Pearson correlation coefficient between the two variables. The Pearson correlation coefficient is a measure of the linear relationship between variables. The closer it is to 1 or −1, the more 'linear' the relationship is. The correlation coefficient can be derived from the following equation:

$$\rho_{x,y} = \frac{N\Sigma X - \Sigma X \Sigma Y}{\sqrt{N\Sigma X^2 - (\Sigma X)^2} * \sqrt{N\Sigma Y^2 - (\Sigma Y)^2}} \tag{1}$$

The Pearson correlation coefficient between crystal size and volume ratio was +0.991184862, which is far greater than +0.8; thus, the correlation was considered strongly linear. We then used linear fitting via Excel to further demonstrate this linear relationship. The errors when reading the lengths of the crystals was considered to be 50 nm. The linear relationship was best demonstrated using the best-fit line in Figure 4. The gradient of the line was 124.05, which describes an increase in size of 124.05 nm every time the ratio increases by 1, starting at ratios greater than or equal to 0.25. The MATLAB code can be found in the Appendix A.

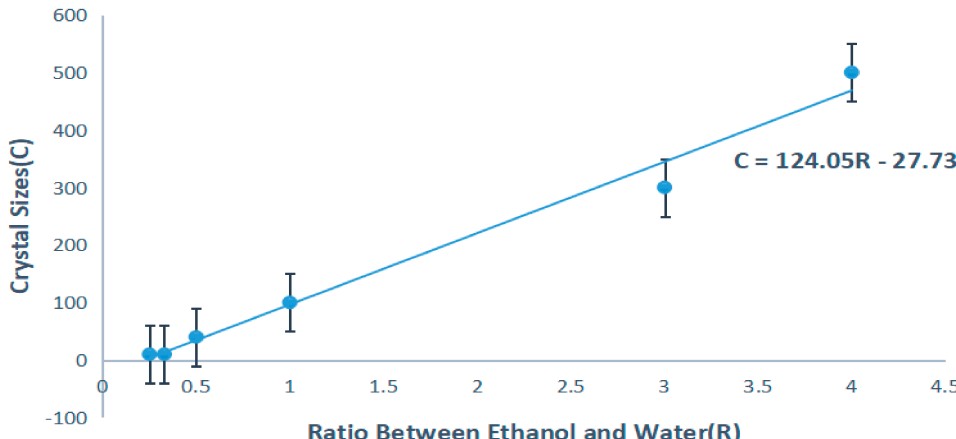

**Figure 4.** The linear relationship between the volume ratio of ethanol to water and crystal size.

Water is known to have the ability to foster the formation of Zr-based metal–organic frameworks. In 2011, Schaate et al. reported that water can increase the rate of the formation of UiO-66, a Zr-based MOF with 12-connected SBUs, straight ditopic linkers, and fcu topology similar to MOF-801. The $OH^-$ ions in water are a key component in the $Zr_6O_4(OH)_4$ SBUs of the 12-connected Zr-based MOF, which consists of both $O^{2-}$ and $OH^-$ ions. With increasing water content, the solution became turbid more quickly, indicating a higher aggregation rate. The increase in the rate of formation of the crystallites signifies a lower possibility for error correction, leading to intergrown structures with low crystallinity. Ethanol, on the other hand, decreased the rate of the solution becoming turbid. The relatively low aggregation rate caused by the increased ratio of ethanol to water is responsible for the bigger single crystals of higher crystallinity. However, what causes the aggregation rate to decrease when the ratio of ethanol to water declines?

To further investigate the role of ethanol in the formation of MOF-801, we applied infrared spectrometry to investigate the specific bonds and functional groups present in the SBU. The black line in Figure 5 demonstrates the infrared spectrum of MOF-801 formed in 20 mL of ethanol and 80 mL of water, named sample (a), while the red line represents that synthesized in 80 mL of ethanol and 20 mL of water, named sample (b). The blue line, on the other hand, represents the FT-IR pattern for the unbounded fumaric acid molecules.

The wavenumbers of some stretching vibration peaks are shown in Table 2. According to the infrared spectra in Figure 5 and the wavenumbers of the peaks in Table 2, the growth mechanism of the MOF-801 of different morphologies can mainly be explained by ethanol's bonding with the Zr metal clusters. We hereby suspect that the ethanol molecules may form coordination bonds with the zirconium ions, blocking the extension of the structure in this specific direction. This is akin to modulating the formation of MOF-801 crystals. Ordinary carboxyl groups, due to the carbonyl groups, exhibit peaks at around 1720 $cm^{-1}$. Furthermore, if no dimers are formed, due to the asymmetric stretch vibration of the O–H, they exhibit a broad peak at around 3400 $cm^{-1}$ [16]. In this case, for the fumaric acid molecules in MOF-801, no dimers were formed. Nevertheless, neither of the IR patterns of these two samples showed peaks around 3550 $cm^{-1}$ or 1720 $cm^{-1}$. They also lacked any consecutive peaks between 2500 $cm^{-1}$ and 3500 $cm^{-1}$. Instead, they had peaks at around 1400 $cm^{-1}$ and 1550 $cm^{-1}$, which signifies the absence of the –COOH group, i.e., the absence of uncoordinated fumaric acid linker molecules, and the exclusive presence of $COO^-$ [10]. The 1400 $cm^{-1}$ peak represents the symmetric stretch vibration of $COO^-$, while the 1550 $cm^{-1}$ peak denotes its asymmetric stretch vibration. According to Dai et al., the IR pattern of the common MOF-801 does not exhibit any strong peaks at around 3400 $cm^{-1}$ [10]. However, there were exceptionally strong absorption peaks in these two samples at around 3400 $cm^{-1}$. Why?

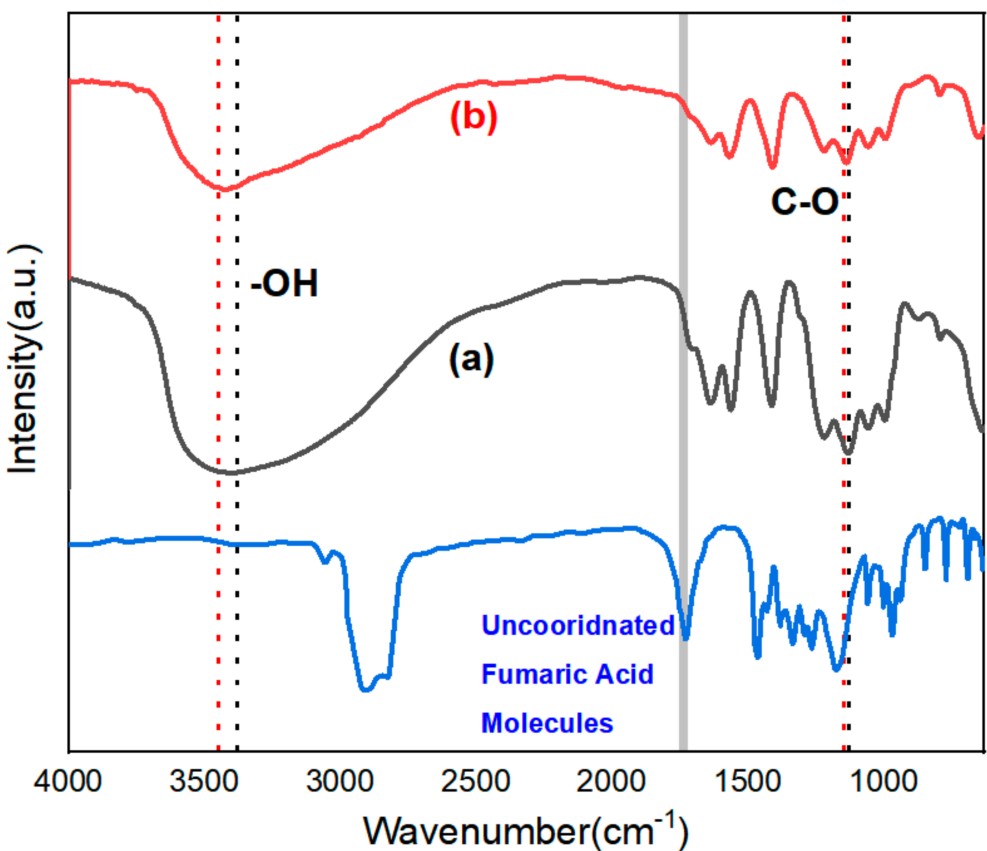

**Figure 5.** The infrared spectra of uncoordinated fumaric acid and MOF-801 formed in (**a**) 80 mL of ethanol, 20 mL of water, and (**b**) 20 mL of ethanol, 80 mL of water.

**Table 2.** The absorption peaks of MOF-801 formed in (a) 80 mL of ethanol, 20 mL of water, and (b) 20 mL of ethanol, 80 mL of water.

| Samples | (a) | (b) |
|---|---|---|
| OH- | 3388.46 cm$^{-1}$ | 3429.72 cm$^{-1}$ |
| COO- | 1412.3 cm$^{-1}$ | 1409.09 cm$^{-1}$ |
| C-OH | 1130.72 cm$^{-1}$ | 1137.37 cm$^{-1}$ |

These strong peaks might have been caused by the hydroxyl groups of ethanol molecules that bonded with the metal nodes [16–18]. The peak for the stretching vibration of the hydroxyl group in ethanol, on the other hand, is at about 3318 cm$^{-1}$. The samples obtained after the reaction in 20 mL of ethanol and 80 mL of water exhibited a peak at 3388 cm$^{-1}$, 78 cm$^{-1}$ higher than the hydroxyl group in ethanol. Similarly, the samples obtained in 80 mL of ethanol and 20 mL of water exhibited a peak at 3429 cm$^{-1}$, 111 cm$^{-1}$ higher than that of the normal hydroxyl group in ethanol. This is because, when ethanol bonds with the Zr(IV) ions, the empty *d*-orbitals of the zirconium ions withdraw electrons from the oxygen atoms of ethanol that are rich in electrons, causing a blue-shift of the absorption peaks for the hydroxyl groups. This blue-shift explains the higher wavelengths absorbed by the hydroxyl groups. A higher proportion of ethanol leads to greater ethanol coordination with the metal ions and, thus, a higher wavenumber.

Ethanol's bonding with the metal ions may also explain the blue-shift of the peak of the C–O bond. The peak for the stretching vibration of the C–O bond in ethanol is at 1038 cm$^{-1}$. When it comes to the as-synthesized sample, the peak shifted to around 1130 cm$^{-1}$, and the intensity of the peak decreased. Due to the electron-withdrawing effect of the Zr ions, a blue-shift of the peak occurred due to an increase in chemical shift [10]. Furthermore,

when the proportion of ethanol increased, the intensities for the peaks corresponding to the C–O bonds and the –OH groups decreased significantly, as shown in Figure 5. This decrease in intensity may be further proof that ethanol coordinates with the metal ions, causing a redistribution of the electrons on the hydroxyl group [19].

We then carried out an XPS narrow-scan analysis to investigate the element content on the surface of the MOF-801 samples. The ratios of oxygen and carbon inside the samples prepared in (a) 80 mL of ethanol, of 20 mL water, and (b) 20 mL of ethanol, 80 mL of water are shown in Figure 6. The percentages of the elements in the samples are shown in Table 3.

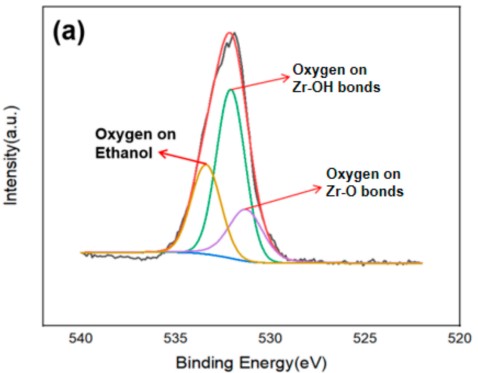 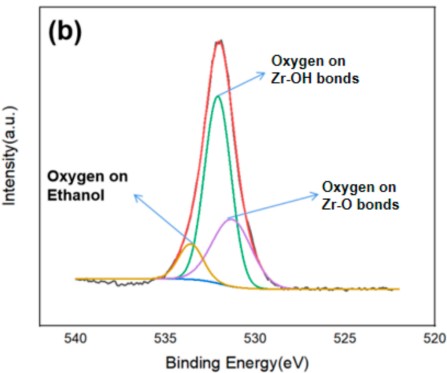

**Figure 6.** The XPS narrow-scan analysis of the oxygen atoms in the MOF-801 samples prepared in (**a**) 80 mL of ethanol, 20 mL of water, and (**b**) 20 mL of ethanol, 80 mL of water.

An interesting finding is that there were three peaks for the $1s$ orbital of oxygen at around 530.4 eV, 531.9 eV and 533.8 eV, which signified the three environments for oxygen atoms. Amongst these three peaks, that at 531.9 eV represents the environment for those oxygen atoms on the Zr–hydroxyl (Zr–OH) bonds inside the SBUs, and that at 530.4 eV is the environment for those oxygen atoms on the Zr–oxo (Zr–O) bonds, including those oxygen atoms from the carboxyl groups of the fumaric acid linkers incorporated in the SBUs of the MOF-801. These peaks are fairly coherent with the previously reported values [20]. Thus, why is there still another peak at 533.8 eV?

We concluded from Figure 5 that there are no unbounded fumaric acid molecules; thus, the only possibility for the other peak is the oxygen atom on ethanol which coordinates with zirconium. We may conclude that the shift in the peak for the hydroxyl group is caused by ethanol molecules bonding with the metal clusters. During the reticulation process when the metal ions cluster together to form stable SBUs, the ethanol molecules also bond with the ions of zirconium due to the strong 'oxophilicity' of zirconium ions. In this way, structural defects emerge on the metal nodes due to the missing linker. Because zirconium ions have empty orbitals and may, thus, withdraw electrons from the hydroxyl groups of the ethanol molecules, a blue-shift of the hydroxyl group is observed. Due to the strength of the Zr–O bonds, the ethanol molecules are not removed in the evacuation step, which washes away the guest molecules residing inside the pores. As indicated by the height of the peaks, when the ratio of the oxygen atoms on ethanol increases (deduced by a height increase in the 533.8 eV peak), the ratio of those on fumaric acid linkers decrease, which agrees with our suggestion.

Another interesting finding was the different percentage ratios of the oxygen and carbon elements. Table 3 further demonstrates that a higher ratio of ethanol led to a larger proportion of carbon and a smaller proportion of oxygen in the as-synthesized sample. The sample synthesized in 20 mL of water and 80 mL of ethanol had 49.01% carbon and only 33.49% oxygen by mass, while that synthesized in 80 mL of water and 20 mL of ethanol contained a smaller proportion of carbon of 33.49% and a higher proportion of oxygen of 45.11%. That is to say, more ethanol led to more carbon atoms and relatively fewer oxygen atoms in the coordination network of MOF-801, which is coherent with our suggestion that ethanol bonds with the metal nodes.

**Table 3.** The percentage ratios of oxygen and carbon in the samples.

| Samples | (a) | (b) |
|---|---|---|
| Oxygen | 33.49% | 45.11% |
| Carbon | 49.01% | 20.32% |

Although the function of ethanol is similar to that of the monotopic modulators, the way in which the ethanol molecules bond with the metal ion clusters is rather different. The differences in their means of bonding are illustrated in Figure 7. As shown in Figure 7, the fumaric acid molecules bond with the metal ion clusters in syn–syn bridging mode, in which two oxygen atoms on a carboxyl group bond with different Zr ions. This is due to the delocalized $\Pi$ bond formed in each of the carbonyl groups, which makes the two oxygen atoms identical and renders the covalent characteristics of the Zr–O bonds. This strategy is called syn–syn bridging [1,14]. This syn–syn bridging strategy explains why the peaks for the linker molecules were around 1400 cm$^{-1}$ and 1550 cm$^{-1}$ rather than 1700 cm$^{-1}$; the two C–O bonds were 'average'. Whereas a fumaric acid molecule is connected to two zirconium ions, two ethanol molecules take the place of this fumaric acid molecule, each bonding with a single zirconium ion. This also constitutes a structure similar to syn–syn bridging. The bonding of these two ethanol molecules blocks the extension of the coordination framework in this specific direction.

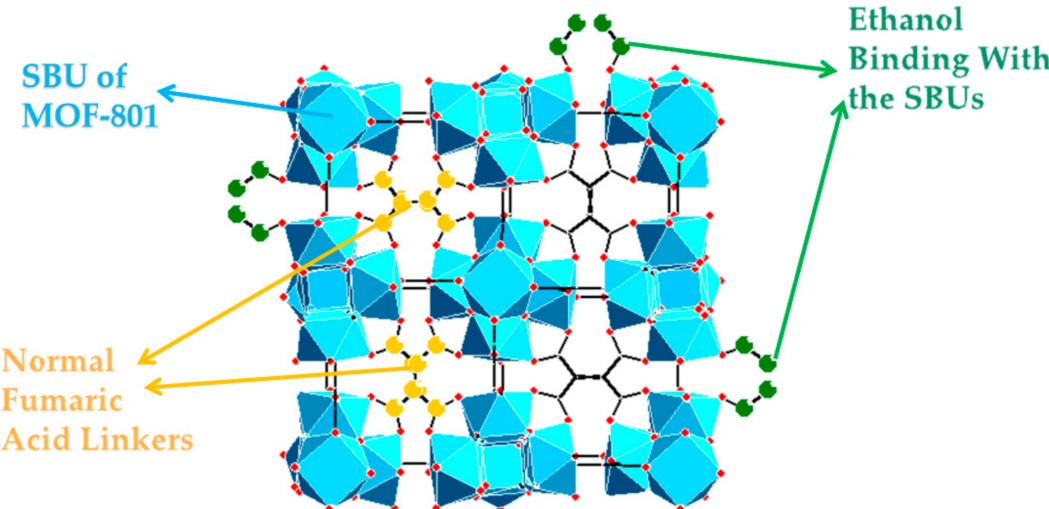

**Figure 7.** The different bonding modes between Zr ions and the organic molecules in MOF-801. All hydrogen atoms are omitted. Both yellow spheres and green atoms represent carbon atoms. They are shown in different colors just for clarification purposes.

What, then, may explain the exception of the MOF-801 nanorod samples? Maybe it is the Marangoni flow effect. The Marangoni flow effect is caused by a surface tension gradient between liquid phases [21–23]. Cai et al. reported that, when a wet nitrogen gas stream passes the surface of a thin layer of ethanol, as the ethanol phase evaporates, water simultaneously condenses near the contact line [12].

Here, as shown in Figure 8, when ethanol evaporates, water molecules begin to gather. Because water evaporates slower than ethanol, the condensed water molecules form a thin water film. Thus, a strong surface tension gradient is generated at the transition zone between the receding ethanol phase and the water phase within a narrow region. The surface tension gradient drives the ethanol, as well as the fumaric acid linker molecules, into the water phase. One-direction flow, therefore, is formed at the interface. When water is added to ethanol, the large difference in surface tension of the liquids immediately drives the flow of ethanol into water. Then, as the reaction proceeds, ethanol constantly evaporates

and condenses, leading to continuous flow of ethanol. This Marangoni flow directs the 1D alignment of the coordination network. The characteristic wavelength shown in Figure 8 can be estimated using Equation (2) [11].

$$\lambda = \frac{2\pi h}{\sqrt{\frac{M_a}{8}}} \tag{2}$$

where $h$ can be calculated from the average diameter of water droplets ($D$) and the contact angle between water droplets and ethanol ($\theta$), as shown in Equation (3).

$$h = \frac{D}{2\sin\theta}(1 - \cos\theta) \tag{3}$$

Ma is the Marangoni number that can be derived from Equation (4).

$$Ma = \frac{\Delta\gamma\Delta T h}{\rho v \kappa} \tag{4}$$

where $\Delta\gamma$ is the difference in surface tension between solvents, $\Delta T$ is the temperature change during reaction, and $\rho v \kappa$ is a constant for water, approximately $1.45933 \times 10^{-8}$ [12]. $D$ is $1.1 \times 10^{-4}$ m, given that a single droplet of water is around 0.05 mL. The difference in surface tension between ethanol (22 mN/m) and water (72 mN/m) is 50 nM/m. Accordingly, the value of $\lambda$ is around 0.764 µm. Thus, when the ratio is 2, the temperature change should be 4.6 °C. This is different from the value observed in Figure 3, which was over 1 µm. What is the reason for this difference?

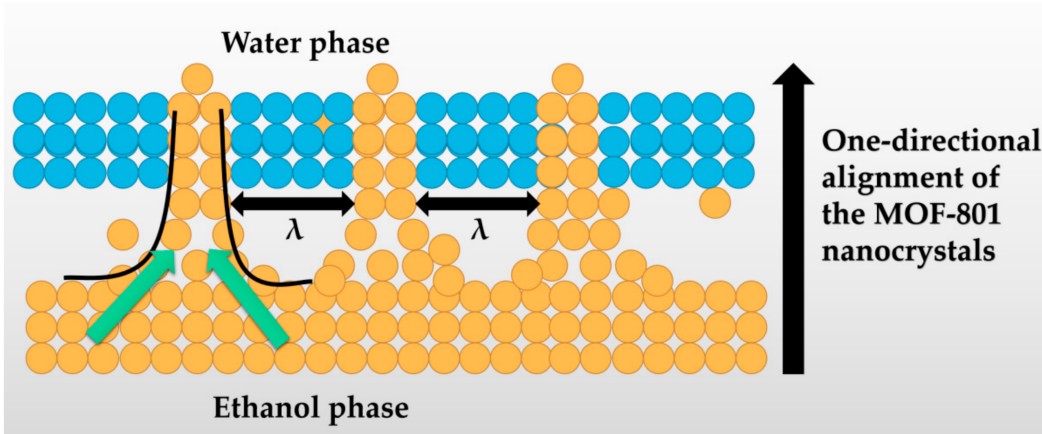

**Figure 8.** Schematic illustration of the Marangoni flow between ethanol and water phase.

Maillard et al. reported that this formula should only be used for quantitative estimations [11]. The difference in surface tension could be less than 50 mN/m due to impurities. There is also thermal energy transfer to the surroundings, which may increase the value of temperature change measured. In this way, the characteristic wavelength of the Marangoni flow could be more than 0.764 µm.

However, what about Marangoni flow for other solvent ratios? It still exists, but it no longer dominates. When the ratio of ethanol to water reaches 3, the temperature change drops to 0.8 °C. This causes the characteristic wavelength to become larger. The MOF-801 single crystals formed in this way are much larger due to larger distances between each stream of the Marangoni convection. On the other hand, when the ratio becomes 1, the temperature change rises to 7.6 °C. Thus, $\lambda$ is much smaller, resulting in relatively smaller sizes of the MOF-801 crystals. These nanosized MOF-801 crystals also easily interconnect because of the smaller distances between each stream of convection. In both of these cases, the role that the Marangoni flow plays no longer dominates.

At this point, we want this work to help further research gain insight into the role that ethanol plays in the synthesis of MOFs. Future research may further investigate the mechanism of how ethanol modulates the reticulation of all sorts of MOFs or focus on new synthesis strategies of MOFs using ethanol.

## 4. Conclusions

The work provides a new strategy for the controllable synthesis of MOF-801 with different morphologies at room temperature. We demonstrate that ethanol, a cost-efficient solvent, can take the place of traditional carboxylic acid modulators due to its competitive coordination to the SBUs with fumaric acid linker molecules. The related mechanism can be well explained by the Marangoni flow effect. Different sizes and morphologies of the MOF-801 can be obtained by simply adjusting the solvent ratio of ethanol to water. This approach may be promising for the controllable synthesis of other types of MOFs with desired morphologies for specified applications.

**Author Contributions:** Conceptualization, Y.L. and C.C.; methodology, Z.T.; project administration, Z.T.; software, Y.L. and C.C.; supervision, C.C.; validation, Y.L.; writing—original draft, Y.L.; writing— review and editing, Y.L., Z.T. and C.C. All authors have read and agreed to the published version of the manuscript.

**Funding:** This research received no external funding.

**Institutional Review Board Statement:** Not applicable.

**Informed Consent Statement:** Not applicable.

**Data Availability Statement:** Please refer to the structure of MOF-801 at https://www.ccdc.cam.ac.uk/ (accessed on 16 April 2021).

**Acknowledgments:** We would like to faithfully thank Tielin Shi for providing the equipment and raw materials. We are also grateful to the Suzhou Institute of Nano-Tech and Nano-Bionics which carried out the XPS, XRD, and BET measurements.

**Conflicts of Interest:** The authors declare no conflict of interest.

## Appendix A

$x$ = [0.25, 0.33, 0.5, 1, 3, 4].
$y$ = [10, 10, 40, 100, 300, 500].
r1 = ($x$, $y$, 'type', 'Pearson').
r2 = corrcoef($x$, $y$).

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
