# Peer review of "The Modulating Effect of Ethanol on the Morphology of a Zr-Based Metal–Organic Framework at Room Temperature in a Cosolvent System"

_crystals, doi:10.3390/cryst11040434_

Round 1

Reviewer 1 Report

This manuscript by Yunzhuo Li, Zirong Tang and Chen Chen report the modulating effect of cosolvent system ethanol:water on the morphology of a Zr-based Metal-Organic Frameworks.

The manuscript presents an accurate introduction to the subject. The authors deeply studied the cosolvent system modulating effect using the characterization techniques of PXRD, FTIR, XPS and SEM. Cosolvent systems are a common and widely used method for obtaining pure and crystalline compounds, particularly among those working in coordination chemistry. It's also true that the attempt mistake is often used to figure out which mixtures perform well on a particular system. The authors present several questions in the text and use a systematic analysis of the data gathered from the characterizations to provide straightforward answers, allowing the text more interesting to read. However, all manuscript seems a little careless when it comes to a fine review and the use of English. There are several loose phrases and the text should be written in past form (past continuous, perfect or perfect continuous). I have no doubt that the authors are aware of what they are writing about, and that the manuscript contains very relevant results in terms of crystal engineering.

The topic of this manuscript fits into the scope of Crystals. I recommend this work for acceptance, although I have some comments and suggestions:

  1. The accuracy of the written unities, must be improved. All units should be separated from its numbers, except for %, e.g. "100mL" should be written "100 mL". When two quantities with the same units are together, only the last one should have the units, simplifying the text reading. Line 95: where it reads: "300 nm and 500 nm are obtained" should be written "300 and 500 nm are obtained”; line 74: "60 degrees Celsius" replace with "60 ºC"; Line 257: unformatted units: "cm-1"; etc. Please review very carefully the "Materials and Methods" topic.
  2. The authors focus on SEM to disclose the particle size, which appears to be inaccurate and simplified. A proper characterization of particle size analysis using traditional techniques with samples dispersed in water is required to support hypothesis presented, such as sentence on line 111.
  3. Perhaps a figure illustrating the reaction, with reagents and the various changed parameters would help to focus the reader's attention.
  4. Loose phrases: An example is found in line 91: "The ratio between ethanol and water is changed to 2." Maybe it's subtitles? Another example follows shortly on line 102.
  5. A lapse on line 208: "cm-1.and"; Also on lines 257 and 258: “cm-1”
  6. Another lapse on line 283: "1-d alignment"
  7. The sentence on line 194 requires clarification. Please feel free to consult the following reference: http://dx.doi.org/10.1063/1.4802991
  8. Latin sentences or words should be italicized “Ceteris Paribus”, " ex-situ", "et al.", etc. Et al. is an abbreviation referring to three Latin expressions that differ only in gender: et alii (masculine plural), et aliae (feminine plural) and et alia (plural neutral); thence I recommend the use of Latin expressions in italic.
  9. Conclusions does not have sufficient arguments to justify the novelty of the present study. Please add a few sentences to justify the novelty of presented study.
  10. What is the take-home message from your study for upcoming MOFs crystallization studies?

Reviewer 2 Report

This paper explores the role of the water/ethanol mixture in the crystallization of MOF-801, using several techniques as SEM, PXRD XPS and IR to characterize the obtained solids.

The manuscript is well written and the results are clearly explained, although some improvements should be made:

-The crystal structure of MOF-801 is cited several times (ie. to simulate the PXRD pattern in figure 2, or in figure 7). The corresponding reference should be inserted.

-Figure 7 is not very illustrative and the quality is low. It should be replaced.

Regarding the PXRD results:

-The experimental conditions used to obtain the patterns should be included. 

-How was the simulated PXRD pattern for MOF-801 generated? It looks like only the positions of the peaks were taken into account.

-A better quality figure for the comparison of PXRD data should be included, either in the manuscript or as supplementary material.

Once this issues have been satisfactorily addressed, I think the manuscript can be accepted for publication.

Round 2

Reviewer 1 Report

This manuscript by Yunzhuo Li, Zirong Tang and Chen Chen report the modulating effect of cosolvent system ethanol:water on the morphology of a Zr-based Metal-Organic Frameworks, MOF-801.

I'd like to thank the authors for responding to all of my questions and comments. It was time and commitment well spent. This article reads much better now, it is clear, has improved, and is ready for publication, but there are a few small errors and some suggestions to improve the manuscript's reading:

  1. Line 73: “60ºC” replace with “60 °C”
  2. Line 81: The authors may improve table 1 by combining the structures of the reagents used, fumaric acid and, Zr(SO4)2, in the reaction to generate the MOF-801. Its formula, [Zr6O4(OH)4(fumarate)6•xH2O]n should also be included in the article, and this is a good spot to do so.
  3. Line 176: Figure 5 can be improved adding the spectra of fumaric acid. As it is, the figure is barely legible.
  4. In figures 1 and 2, how were the MOF-801 diffractograms simulated? I didn't see any mention to single crystals. I was left with this critical question.
